# Clustering with Bregman Divergences: an Asymptotic Analysis

**Chaoyue Liu, Mikhail Belkin**
Department of Computer Science & Engineering
The Ohio State University
liu.2656@osu.edu, mbelkin@cse.ohio-state.edu

## Abstract

Clustering, in particular $k$-means clustering, is a central topic in data analysis. Clustering with Bregman divergences is a recently proposed generalization of $k$-means clustering which has already been widely used in applications. In this paper we analyze theoretical properties of Bregman clustering when the number of the clusters $k$ is large. We establish quantization rates and describe the limiting distribution of the centers as $k \to \infty$, extending well-known results for $k$-means clustering.

## 1 Introduction

Clustering and the closely related problem of vector quantization are fundamental problems in machine learning and data mining. The aim is to partition similar points into "clusters" in order to organize or compress the data. In many clustering methods these clusters are represented by their centers or centroids. The set of these centers is often called "the codebook" in the vector quantization literature. In this setting the goal of clustering is to find an optimal codebook, i.e., a set of centers which minimizes a clustering loss function also known as the quantization error.

There is vast literature on clustering and vector quantization, see, e.g., [8, 10, 12]. One of the particularly important types of clustering and, arguably, of data analysis methods of any type, is $k$-means clustering [16] which aims to minimize the loss function based on the squared Euclidean distance. This is typically performed using the Lloyd's algorithm [15], which is an iterative optimization technique. The Lloyd's algorithm is simple, easy to implement and is guaranteed to converge in a finite number of steps. There is an extensive literature on various aspects and properties of $k$-means clustering, including applications and theoretical analysis [2, 13, 23]. An important type of analysis is the asymptotic analysis, which studies the setting when the number of centers is large. This situation ($n \gg k \gg 0$) arises in many applications related to data compression as well as algorithms such as soft $k$-means features used in computer vision and other applications, where the number of centers $k$ is quite large but significantly less than the number of data points $n$. This situation also arises in $k$-means feature-based methods which have seen significant success in computer vision, e.g., [6]. The quantization loss for $k$-means clustering in the setting $k \to \infty$ is well-known (see [5, 9, 20]). A less well-known fact shown in [9, 18] is that the discrete set of centers also converges to a measure closely related to the underlying probability distribution. This fact can be used to reinterpret $k$-means feature based methods in terms of a density dependent kernel [21].

More recently, it has been realized that the properties of square Euclidean distance which make the Lloyd's algorithm for $k$-means clustering so simple and efficient are shared by a class of similarity measures based on Bregman divergence. In an influential paper [3] the authors introduced clustering based on Bregman divergences, which generalized $k$-means clustering to that setting and produced a corresponding generalized version of the Lloyd's algorithm. That work has lead to a new line of research on clustering including results on multitask Bregman clustering[24], agglomerative

Bregman clustering[22] and many others. There has also been some theoretical analysis of Bregman clustering including [7] proving the existence of an optimal quantizer and convergence and bounds for quantization loss in the limit of size of data $n \to \infty$ for fixed $k$.

In this paper we set out to investigate asymptotic properties of Bregman clustering as the number of centers increases. We provide explicit asymptotic rates for the quantization error of Bregman clustering as well as the continuous measure which is the limit of the center distribution. Our results generalize the well-known results for $k$-means clustering. We believe that these results will be useful for better understanding in Bregman divergence based clustering algorithms and algorithms design.

## 2 Preliminaries and Existing Work

### 2.1 $k$-means clustering and its asymptotic analysis

$k$-means clustering is one of the most popular and well studied clustering problems in data analysis.

Suppose we are given a dataset $\mathcal{D} = \{x_i\}_{i=1}^n \subset \mathbb{R}^d$ , containing $n$ observations of a $\mathbb{R}^d$-valued random variable $X$. $k$-means clustering aims to find a set of points (centroids) $\alpha = \{a_j\}_{j=1}^k \subset \mathbb{R}^d$, with $|\alpha| = k$ initially fixed, that minimizes the squared Euclidean loss function

$$L(\alpha) = \frac{1}{n} \sum_j \min_{a \in \alpha} \|x_j - a\|_2^2. \tag{1}$$

Finding the global minimum of loss function is a NP-hard problem [1, 17]. However, Lloyd's algorithm [15] is a simple and elegant method to obtain a locally optimal clustering of the data, corresponding to a local minimum of the loss function. A key reason for the practical utility of the Lloyd's $k$-means algorithm is the following property of squared Euclidean distance: the arithmetic mean of a set of points is the unique minimizer of the loss for a single center:

$$\frac{1}{n} \sum_{i=1}^n x_i = \arg \min_{s \in \mathbb{R}^d} \frac{1}{n} \sum_{i=1}^n \|x_i - s\|_2^2. \tag{2}$$

It turns out that this property holds in far greater generality as we will discuss below.

**Asymptotic analysis of Euclidean quantization:**

In an asymptotic quantization problem, we focus on the limiting case of $k \to \infty$, where the size of dataset $n \gg k$. In this paper we will assume $n = \infty$, i.e., that the probability distribution with density $P$ is given. This setting is in line with the analysis in [9].

Correspondingly, a density measure $\mathcal{P}$ is defined as follows: for a set $A \subseteq \mathbb{R}^d$, $\mathcal{P}(A) = \int_A P d\lambda^d$, where $\lambda^d$ is the Lebesgue measure on $\mathbb{R}^d$. We also have $P = \frac{d\mathcal{P}}{d\lambda^d}$.

The classical asymptotic results for the Euclidean quantization are provided in a more general setting for an arbitrary power of the distance Eq.(1), Euclidean quantization of order $r$ ($1 \le r < \infty$), with loss

$$L(\alpha) = E_P \left[ \min_{a \in \alpha} \|X - a\|_2^r \right]. \tag{3}$$

Note that the Lloyd's algorithm is only applicable to the standard case with $r = 2$.

The output of the $k$-means algorithm include locations of centroids, which then imply the partition and the corresponding loss. For large $k$ we are interested in: (1) *the asymptotic quantization error*, and (2) *the distribution of centroids*.

**Asymptotic quantization error.** The asymptotic quantization error for $k$-means clustering has been analyzed in detail in [5, 14, 20]. S. Graf and H. Luschgy [9] show that as $k \to \infty$, the $r$-th quantization error decreases at a rate of $k^{-r/d}$. Furthermore, coefficient of the term $k^{-r/d}$ is of the form

$$Q_r(P) = Q_r([0,1]^d) \|P\|_{d/(d+r)}, \tag{4}$$

where $Q_r([0,1]^d)$, a constant independent of $P$, is geometrically interpreted as asymptotic Euclidean quantization error for uniform distribution on $d$-dimensional unite cube $[0,1]^d$. Here $\|\cdot\|_{d/(d+r)}$ is the $L^{d/(d+r)}$ norm of function: $\|f\|_{d/(d+r)} = (\int f^{d/(d+r)} d\lambda^d)^{(d+r)/d}$.

**Locational distribution of centroids.** A less well-known fact is that the locations of the optimal centroid configuration of $k$-means converges to a limit distribution closely related to the underlying density [9, 18]. Specifically, given a discrete set of centroids $\alpha_k$, to construct the corresponding discrete measure,

$$\mathcal{P}_k = \frac{1}{k} \sum_{j=1}^{k} \delta_{a_j}, \tag{5}$$

where $\delta_a$ is Dirac measure centered at $a$. For a open set $A \subseteq \mathbb{R}^d$, $\mathcal{P}_k(A)$ is the ratio of number of centroids $k_A$ located within $A$ to the total number of centroids $k$, namely $\mathcal{P}_k(A) = k_A/k$. We say that a continuous measure $\tilde{\mathcal{P}}$ is the limit distribution of centroids, if $\{\mathcal{P}_k\}$ (weakly) converges to $\tilde{\mathcal{P}}$, specifically

$$\forall A \subseteq \mathbb{R}^d, \lim_{k \to \infty} \mathcal{P}_k(A) = \tilde{\mathcal{P}}(A). \tag{6}$$

S. Graf and H. Luschgy [9] gave an explicit expression for this continuous limit distribution of centroids:

$$\tilde{\mathcal{P}}_r = \tilde{P}_r \lambda^d, \quad \tilde{P}_r = \mathcal{N} \cdot P^{d/(d+r)}, \tag{7}$$

where $\lambda^d$ is the Lebesgue measure on $\mathbb{R}^d$, $P$ is the density of the probability distribution and $\mathcal{N}$ is the normalization constant to make sure that $\tilde{P}_r$ integrates to 1.

## 2.2 Bregman divergences and Bregman Clustering

In this section we briefly review basics of Bregman divergences and the Bregman clustering algorithm.

Bregman divergence, first proposed in 1967 by L.M.Bregman [4], measure dissimilarity between two points in a space. The formal definition is as follows:

**Definition 1** (Bregman Divergence). *Let function $\phi$ be strictly convex on a convex set $\Omega \subseteq \mathbb{R}^d$, such that $\phi$ is differentiable on relative interior of $\Omega$, we define Bregman divergence $D_\phi : \Omega \times \Omega \to \mathbb{R}$ with respect to $\phi$ as:*

$$D_\phi(p, q) = \phi(p) - \phi(q) - \langle p - q, \nabla\phi(q) \rangle, \tag{8}$$

*where $\langle \cdot, \cdot \rangle$ is inner product in $\mathbb{R}^d$. $\Omega$ is domain of the Bregman divergence.*

Note that Bregman divergences are not necessarily true metrics. In general, they do satisfy the basic properties of non-negativity and identity of indiscernibles, but may not respect the triangle inequality and symmetry.

**Examples:** Some popular examples of Bregman divergences include:

Squared Euclidean distance: $\quad D_{EU}(p, q) = \|p - q\|_2^2, \quad (\phi_{EU}(z) = \|z\|^2)$

Mahalanobis distance: $\quad D_{MH}(p, q) = (p - q)^T A(p - q), \quad A \in \mathbb{R}^{d \times d}$

Kullback-Leibler divergence: $\quad KL(p\|q) = \sum p_i \ln \frac{p_i}{q_i} - \sum (p_i - q_i),$

$$(\phi_{KL}(z) = \sum z_i \ln z_i - z_i, \quad z_i > 0)$$

Itakura-Saito divergence: $\quad D_{IS}(p\|q) = \sum \frac{p_i}{q_i} - \ln \frac{p_i}{q_i} - 1, \quad (\phi_{IS}(z) = -\sum \ln z_i)$

Norm-like divergence: $\quad D_{NL}(p\|q) = \sum_i p_i^\alpha + (\alpha - 1)q_i^\alpha - \alpha p_i q_i^{\alpha-1}.$

$$(\phi_{NL}(z) = \sum z_i^\alpha, \quad z_i > 0, \alpha \geq 2) \tag{9}$$

Domains of Bregman divergences: $\Omega_{EU} = \Omega_{MH} = \mathbb{R}^d$, and $\Omega_{KL} = \Omega_{IS} = \Omega_{NL} = \mathbb{R}_+^d$.

**Alternative expression: the quadratic form.** Suppose that $\phi \in C^2(\Omega)$, which holds for most popularly used Bregman divergences. Note that $\phi(q) + \langle p - q, \nabla\phi(q) \rangle$ is simply the first two terms in Taylor expansion of $\phi$ at $q$. Thus, Bregman divergences are nothing but the difference between a function and its linear approximation. By Lagrange's form of the remainder term, there exists $\xi$ with $\xi_i \in [\min(p_i, q_i), \max(p_i, q_i)]$ (i.e. $\xi$ is in the smallest d-dimensional axis-parallel cube that contains $p$ and $q$) such that

$$D_\phi(p, q) = \frac{1}{2}(p - q)^T \nabla^2\phi(\xi)(p - q), \tag{10}$$

where $\nabla^2 \phi(\xi)$ denotes the Hessian matrix of $\phi$ at $\xi$.

This form is more compact and will be convenient for further analysis, but at the expense of introducing an unknown point $\xi$. We will use this form in later discussions.

**The mean as the minimizer.** As shown in A. Banerjee et al. [3], the property Eq.(2) still holds if squared Euclidean distance is substituted by a general Bregman divergence:

$$\frac{1}{n}\sum_{i=1}^{n} x_i = \arg\min_{s \in \Omega} \sum_{i=1}^{n} D_\phi(x_i, s). \tag{11}$$

That allows for the Lloyd's method to be generalized to arbitrary Bregman clustering problems, where the new loss function is defined as

$$L(\alpha) = \frac{1}{n}\sum_i \min_{a \in \alpha} D_\phi(x_i, a). \tag{12}$$

This modified version of $k$-means, called Bregman hard clustering algorithm (see Algorithm 1 in [3]), results a locally optimal quantization as well.

## 3 Asymptotic Analysis of Bregman Quantization

We do not distinguish the terminology of Bregman quantization and Bregman clustering. In this section, we analyze the asymptotics of Bregman quantization allowing a power of Bregman divergences in the loss function. We show expressions for the quantization error and limiting distribution of centers.

We start with the following:

**Definition 2** (k-th quantization error for P of order r)**.** *Suppose a variable $X$ takes values on $\Omega \subseteq \mathbb{R}^d$ following a density $P$, where $\Omega$ is the $d$-dimensional domain of Bregman divergence $D_\phi$. The k-th quantization error for P of order r ($1/2 \leq r < \infty$) associated with $D_\phi$ is defined as*

$$V_{k,r,\phi}(P) = \inf_{\alpha \subset \mathbb{R}^d, |\alpha|=k} E_P\left[\min_{a \in \alpha} D_\phi^r(X, a)\right] \tag{13}$$

*where $\alpha \subset \mathbb{R}^d$ is set of representatives of clusters, corresponding to a certain partition, or quantization of $\mathbb{R}^d$ or support of $P$, and $E_P[\cdot]$ means taking expectation value over $P$.*

**Remark:** **(a)** The set $\alpha^*$ that reaches the infimum is called *k-optimal set of centers* for $P$ of order $r$ with respect to $D_\phi^r(X, a)$. **(b)** In this setting, Bregman quantization of order $r$ corresponds to Euclidean quantization of order $2r$, because of Eq.(10).

### 3.1 Asymptotic Bregman quantization error

We are interested in the asymptotic case, where $k \to \infty$.

First note that quantization error asymptotically approaches zero as every point $x$ in the support support of the distribution can always is arbitrarily closed to a centroid with respect to the Bregman divergence when $k$ is large enough.

**Intuition on Convergence rate.** We start by providing an informal intuition for the convergence rate. Assume $P$ has a compact support with a finite volume. Suppose each cluster is a (Bregman) Voronoi cell with typical size $\epsilon$. Since total volume of the support does not change, volume of one cell should be inversely proportional to the number of clusters, $\epsilon^d \sim \frac{1}{k}$. On the other hand, because of Eq.(10), Bregman divergence between two points in one cell is the order of square of the cell size, $D_\phi(X, a) \sim \epsilon^2$, That implies

$$V_{k,r,\phi}(P) \sim k^{-2r/d} \text{ asymptotically.} \tag{14}$$

We will now focus making this intuition precise and on deriving an expression for the coefficient at the leading term $k^{-2r/d}$ in the quantization error. For now we will keep the assumption that $P$ has compact support, and remove it later on. We only describe the method and display important results in the following. Please see detailed proofs of these results in the Appendix.

We first mention a few useful facts:

**Lemma 1.** *In the limit of $k \to \infty$, each interior point $x$ in the support of $P$ is assigned to an arbitrarily close centroid in the optimal Bregman quantization setting.*

**Lemma 2.** *If support of $P$ is convex, $\phi$ is strictly convex on the support and $\nabla^2 \phi$ is uniformly continuous on the support, then* (a): $\lim_{k \to \infty} k^{\frac{2r}{d}} V_{k,r,\phi}(P)$ *exists in $(0, \infty)$, denoted as $Q_{r,\phi}(P)$, and* (b):

$$Q_{r,\phi}(P) = \lim_{k \to \infty} k^{\frac{2r}{d}} \inf_{\alpha(|\alpha|=k)} E_P \left[ \min_{a \in \alpha} \left( \frac{1}{2}(X-a)^T \nabla^2\phi(a)(X-a) \right)^r \right]. \tag{15}$$

**Remark: 1,** Since $Q_{r,\phi}(P)$ is finite, part (a) of Lemma 2 proves our intuition on convergence rate, Eq.(14). **2,** In Eq.(15), it does not matter whether $\nabla^2\phi$ take values at $a$, $x$ or even any point between $x$ and $a$, as long as $\nabla^2\phi$ has finite values at that point.

**Coefficient of Bregman quantization error.** We evaluate the coefficient of quantization error $Q_{r,\phi}(P)$, based on Eq.(15). What makes this analysis challenging is that unlike is that Euclidean quantization, general Bregman error does not satisfy translational invariance and scaling properties. For example, Lemma 3.2 in [9] does not hold for general Bregman divergence. We follow the following approach: First, dice the the support of $P$ into infinitesimal cubes $\{A_l\}$ with edges parallel to axes, where $l$ is the index for cells. In each cell, we approximate the Hessian by a constant matrix $\nabla^2\phi(z_l)$, where $z_l$ is a fixed point located in the cell. Therefore, evaluating the Bregman quantization error within each cell reduces to a Euclidean quantization problem, with existing result, Eq.(4). Then summing them up appropriately over the cubes gives total quantization error.

We start from Eq.(15), and introduce the following notation: denote $s_l = \mathcal{P}(A_l)$ and conditional density on $A_l$ as $P(\cdot|A_l)$, $\alpha_l = \alpha \cap A_l$ as set of centroids that located in $A_l$ and $k_l = |\alpha_l|$ as size of $\alpha_l$, and ratio $v_l = k_l/k$. Following the above intuition and noting that $P = \sum \mathcal{P}(A_l) P(\cdot|A_l)$, $Q_{r,\phi}(P)$ is approximated by

$$Q_{r,\phi}(P, \{v_l\}) \quad \sim \quad \sum_l s_l v_l^{-2r/d} Q_{r,Mh,l} \left( P(\cdot|A_l) \right), \tag{16}$$

$$Q_{r,Mh,l} \left( P(\cdot|A_l) \right) \quad = \quad \lim_{k_l \to \infty} k_l^{\frac{2r}{d}} \inf_{\alpha_l(|\alpha_l|=k_l)} E_{P(\cdot|A_l)} \left[ \min_{a \in \alpha_l} \frac{1}{2}(X-a)^T \nabla^2\phi(z_l)(X-a) \right]^r \tag{17}$$

where $Q_{r,Mh,l} \left( P(\cdot|A_l) \right)$ is coefficient of asymptotic Mahalanobis quantization error with Mahalanobis matrix $\nabla^2\phi(z_l)$, evaluated on $A_l$ with density $P(\cdot|A_l)$. It can be shown that the approximation error of $Q_{r,\phi}(P)$ converges to zero in the limits of $k \to \infty$ and then size of cell to zero.

In each cell $A_l$, $P(\cdot|A_l)$ is further approximated by uniform density $U(A_l) = 1/V_l$, and Hessian $\nabla^2\phi(z_l)$, as a constant, is absorbed by performing a coordinate transformation. Then $Q_{r,Mh,l} \left( U(A_l) \right)$ reduces to squared Euclidean quantization error. By applying Eq.(4), we show that

$$Q_{r,Mh,l} \left( U(A_l) \right) = \frac{1}{2^r} Q_{2r}([0,1]^d) \delta^{2r} [\det \nabla^2\phi(z_l)]^{r/d} \tag{18}$$

where $\delta$ is the size of cube, and $Q_{2r}([0,1]^d)$ is again the constant in Eq.(4).

Combining Eq.(17) and Eq.(18), $Q_{r,\phi}(P)$ is approximated by

$$Q_{r,\phi}(P, \{v_l\}) \sim \frac{1}{2^r} Q_{2r}([0,1]^d) \delta^{2r} \sum_l s_l v_l^{-2r/d} [\det \nabla^2\phi(z_l)]^{r/d}. \tag{19}$$

Portion of centroids $v_l$ within $A_l$ is still undecided yet. The following lemma provides an optimal configuration of $\{v_l\}$ that minimizes $Q_{r,\phi}(P, \{v_l\})$:

**Lemma 3.** *Let $B = \{(v_1, \cdots, v_L) \in (0,\infty)^L : \sum_{l=1}^L v_l = 1\}$, and define*

$$v_l^* = \frac{s_l^{d/(d+2r)} [\det \nabla^2\phi(z_l)]^{r/(d+2r)}}{\sum_l s_l^{d/(d+2r)} [\det \nabla^2\phi(z_l)]^{r/(d+2r)}}, \tag{20}$$

*then for the function*

$$F(v_1, \cdots, v_L) = \sum_{l=1}^L s_l v_l^{-2r/d} [\det \nabla^2\phi(z_l)]^{r/d}, \tag{21}$$

$$F(v_1^*, \cdots, v_L^*) = \min_{(v_1, \cdots, v_L) \in B} F(v_1, \cdots, v_L) = \left( \sum_l s_l^{d/(d+2r)} [\det \nabla^2 \phi(z_l)]^{r/(d+2r)} \right)^{(d+2r)/d}.$$

(22)

Lemma 3 finds the optimal configuration of $\{v_l\}$ in Eq.(19). Recall that quantization error is defined to be infimum over all possible configurations, we have our main result:

**Theorem 1.** *Suppose $E||X||^{2r+\epsilon} < \infty$ for some $\epsilon > 0$ and $\nabla^2(\phi)$ is uniformly continuous on the support of P, then*

$$Q_{r,\phi}(P) = \frac{1}{2^r} Q_{2r}([0,1]^d) \|(\det \nabla^2 \phi)^{r/d} P\|_{d/(d+2r)}.$$

(23)

**Remark: 1,** In the Euclidean quantization cases, where $\phi(z) = ||z||^2$, Eq.(23) reduces to Eq.(4), noting that $\nabla^2 \phi = 2\mathbf{I}$. Bregman quantization, which is more general than Euclidean quantization, has result that is quite similar to Eq.(4), differing by a $\det \nabla^2 \phi$-related term.

### 3.2 The Limit Distribution of Centroids

Similar to Euclidean clustering, Bregman clustering also outputs $k$ discrete cluster centroids, which can be interpreted as a discrete measure. Below we show that in the limit this discrete measure coincide with a continuous measure defined in terms of the probability density $P$.

Define $P_{r,\phi}$ to be the integrand function in Eq.(23) (with a normalization factor $\mathcal{N}$),

$$P_{r,\phi} = \mathcal{N} \cdot (\det \nabla^2 \phi)^{r/(d+2r)} P^{d/(d+2r)}.$$

(24)

The following theorem claim that $P_{r,\phi}$ is exactly the continuous distribution we are looking for:

**Theorem 2.** *Suppose $\mathcal{P}$ is absolutely continuous with respect to Lebesgue measure $\lambda^d$. Let $\alpha_k$ be an asymptotically k-optimal set of centers for P of order r based on $D_\phi$. Define measure $\mathcal{P}_{r,\phi} := P_{r,\phi} \lambda^d$, then*

$$\frac{1}{k} \sum_{a \in \alpha_k} \delta_a \to \mathcal{P}_{r,\phi} \text{ (weakly)}.$$

(25)

**Remark:** As before $\mathcal{P}_{r,\phi}$ is the measure while $P_{r,\phi}$ is the corresponding density function. The proof of the theorem can be found in the appendix.

**Example 1: Clustering with Squared Euclidean distance (Graf and Luschgy[9]).** Squared Euclidean distance is an instance of Bregman divergence, with $\phi(z) = \sum z_i^2$. Graf and Luschgy proved that asymptotic centroid's distribution is like

$$P_{r,EU}(z) \sim P^{d/(d+2r)}(z).$$

(26)

**Example 2: Clustering with Mahalanobis distance.** Mahalanobis distance is another instance of Bregman divergence, with $\phi(z) = z^T A z, (A) \in \mathbb{R}^d$. Hessian matrix $\nabla^2 \phi = A$. Then the asymptotic centroid's distribution is same as that of Squared Euclidean distance

$$P_{r,Mh}(z) \sim P^{d/(d+2r)}(z).$$

(27)

**Example 3: Clustering with Kullback-Leibler divergence.** The convex function used to define Kullback-Leibler divergence is negative Shannon entropy defined on domain $\Omega \subseteq \mathbb{R}_+^d$,

$$\phi_{KL}(z) = \sum_i z_i \ln z_i - z_i$$

(28)

with component index $i$. Then Hessian matrix

$$\nabla^2 \phi_{KL}(z) = \text{diag}(\frac{1}{z_1}, \frac{1}{z_2}, \cdots, \frac{1}{z_d}).$$

(29)

According to Eq. (24), centroid's density distribution function

$$P_{r,KL}(z) \sim P^{d/(d+2r)}(z) \left( \prod_i z_i \right)^{-r/(d+2r)} . \tag{30}$$

**Example 4: Clustering with Itakura-Saito divergence.** Itakura-Saito divergence uses Burg entropy as $\phi$,

$$\phi_{IS}(z) = -\sum_i \ln z_i, \quad z \in \mathbb{R}^d, \tag{31}$$

with component index $i$. Then Hessian matrix

$$\nabla^2 \phi_{IS}(z) = \text{diag}(\frac{1}{z_1^2}, \frac{1}{z_2^2}, \cdots, \frac{1}{z_d^2}). \tag{32}$$

According to Eq. (24), centroid's density distribution function

$$P_{r,IS}(z) \sim P^{d/(d+2r)}(z) \left( \prod_i z_i^2 \right)^{-r/(d+2r)} . \tag{33}$$

**Example 5: Clustering with Norm-like divergence.** Convex function $\phi(z) = \sum_i z_i^\alpha, z \in \mathbb{R}_+^d$, with power $\alpha \geq 2$. Simple calculation shows that the divergence reduces to Euclidean distance when $\alpha = 2$. However, the divergence is no longer Euclidean-like, as long as $\alpha > 2$:

$$D_{NL}(p,q) = \sum_i p_i^\alpha + (\alpha - 1)q_i^\alpha - \alpha p_i q_i^{\alpha-1}. \tag{34}$$

With some calculation, we have

$$P_{r,NL}(z) \sim P^{d/(d+2r)}(z) \left( \prod_i z_i \right)^{(\alpha-2)r/(d+2r)} . \tag{35}$$

**Remark:** It is easy to see that Kullback-Leibler and Itakura-Saito quantization tend to move centroids closer to axes, and Norm-like quantization, when $\alpha > 2$, does opposite thing, moving centroids far away from axes.

## 4 Experiments

In this section, we verify our results, especially centroid's location distribution Eq.(24), by using the Bregman hard clustering algorithm.

Recall that our results are obtained in a limiting case, where we first take size of dataset $n \to \infty$ and then number of clusters $k \to \infty$. However, size of real data is finite and it is also not practical to apply Bregman clustering algorithms on the asymptotic case. In this section, we simply sample data points from given distribution, with dataset size large enough, compared to $k$, to avoid early stopping of Bregman clustering. In addition, we only verify $r = 1$ cases here, since the Bregman clustering algorithm, which utilizes Lloyd's method, cannot address Bregman quantization problems with $r \neq 1$.

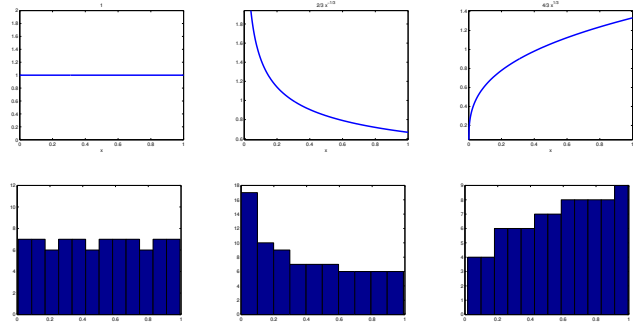

Squared Euclidean    Kullback-Leibler    Norm-like ($\alpha = 3$)

Figure 1: First row are predicted distribution functions of centroids by Eq.(36,37,38); second row are experimental histograms of location of centroids, by applying corresponding Bregman hard clustering algorithms.

**Case 1 (1-dimensional):** Suppose the density $P$ is uniform over $[0, 1]$. We set number of clusters $k = 81$, and apply different versions of Bregman hard clustering algorithm on this sample: standard $k$-means, Kullback-Leibler clustering and norm-like clustering. According to Eq.(27), Eq.(33) and Eq.(35), theoretical prediction of centroids locational distribution functions in this case should be:

$$P_{1,EU}(z) = 1, \quad z \in [0, 1]; \tag{36}$$

$$P_{1,KL}(z) \sim z^{-1/3}, \quad z \in (0, 1]; \tag{37}$$

$$P_{1,NL}(z) \sim z^{1/3}, \quad z \in [0, 1]; \tag{38}$$

and $P(z) = 0$ elsewhere.

Figure 1 shows, in the first row, the theoretical prediction of distribution of centroids, and in the second row, experimental histograms of centroid locations for different Bregman quantization problems.

**Case 2 (2-dimensional):** The density $P = U([0, 1]^2)$. Set $k = 81$ and apply the same three Bregman clustering algorithms as in case 1. Theoretical predictions of distribution of centroids for this case by Eq.(27), Eq.(33) and Eq.(35) are as follow, also shown in Figure 2:

$$P_{1,EU}(z) = 1, \quad z = (z_1, z_2) \in [0, 1]^2; \tag{39}$$

$$P_{1,KL}(z) \sim (z_1 z_2)^{-1/4}, \quad z = (z_1, z_2) \in (0, 1]^2; \tag{40}$$

$$P_{1,NL}(z) \sim (z_1 z_2)^{1/4}, \quad z = (z_1, z_2) \in [0, 1]^2; \tag{41}$$

and $P(z) = 0$ elsewhere.

Figure 2, in the first row, shows a visualization of centroids locations generated by experiments. For comparison, second row of Figure 2 presents 3-d plots of theoretical predictions of distribution of centroids. In each of the 3-d plots, function is plotted over the cube $[0, 1]^2$, with left most corner corresponding to point $(0, 0)$.

It is easy to see that squared Euclidean quantization, in this case, results an uniform distribution of centroids, and that Kullback-Leibler quantization tends to attract centroids towards axes, and norm-like quantization tends repel centroids away from axes.

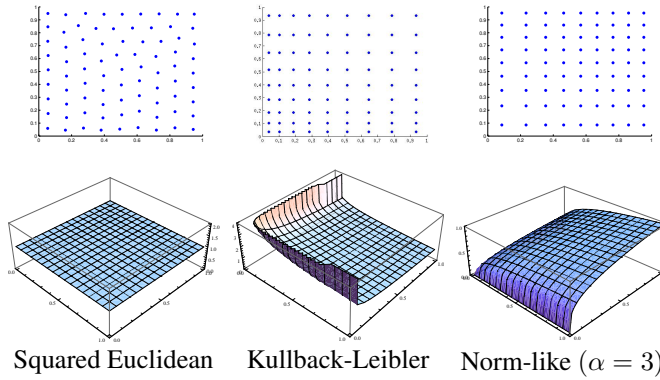

Squared Euclidean     Kullback-Leibler     Norm-like $(\alpha = 3)$

Figure 2: Experimental results and theoretical predictions of centroids distribution for Case 2. In each of the 3-d plots, function is plotted over the cube $[0, 1]^2$, with left most corner corresponding to point $(0, 0)$, and right most corner corresponding to point $(1, 1)$.

## 5 Conclusion

In this paper, we analyzed the asymptotic Bregman quantization problems for general Bregman divergences. We obtained explicit expressions for both leading order of asymptotic quantization error and locational distribution of centroids, both of which extend the classical results for $k$-means quantization. We showed how our results apply to commonly used Bregman divergences, and gave some experimental verification. We hope these results will provide guidance and insight for further theoretical analysis of Bregman clustering, such as Bregman soft clustering and other related methods [3, 11], as well as for practical algorithm design and applications. Our results can also lead to better understanding of the existing seeding strategies for Bregman clustering [19] and to new seeding methods.

**Acknowledgement**

We thank the National Science Foundation for financial support and to Brian Kulis for discussions.

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
