[Supplementary Material]

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

# Appendix

## Proof of Lemma 1:

*Proof. Claim 1:* For any two interior points $x, a \in \Omega$, $x \neq a$, $D_\phi(x, a) > 0$.

Let's start from Eq.(10). Since $\Omega$ is convex, and $\xi$ is a point between $x$ and $a$, $\xi$ is an interior point in $\Omega$. Because $\phi$ is strictly convex on $\Omega$ by definition, Hessian matrix $\nabla^2 \phi(\xi)$ is positive definite. Then $x - a \neq 0$ implies $D_\phi(x, a) > 0$.

*Claim 2:* $\forall \epsilon > 0, \exists a \in \Omega$ s.t. $D_\phi(x, a) < \epsilon$.

For the interior point $x$, we can always find a $d$-dimensional ball $B(x, \delta) \subset \Omega \subseteq \mathbb{R}^d$, such that $\nabla^2 \phi$ is upper bounded by some finite $\Lambda > 0$ in the following sense:

$$\forall v \in \mathbb{R}^d, \forall \xi \in B(x, \delta), v^T \nabla^2 \phi(\xi) v \leq \Lambda \|v\|^2. \tag{42}$$

Set $\delta' = \min\{\sqrt{\epsilon/\Lambda}, \delta\} > 0$, then

$$\forall y \in B(x, \delta'), D_\phi(x, y) = (x - y)^T \nabla^2 \phi(\xi)(x - y) < \Lambda \|x - y\|^2 \leq \Lambda \delta'^2 \leq \epsilon. \tag{43}$$

And we also have $\alpha \cap B(x, \delta') \neq \emptyset$ in the limit of $k \to \infty$. This is because, if $\alpha \bigcap B(x, \delta') = \emptyset$, adding one centroid in $B(x, \delta')$ would decrease the quantization error for sure. Thus, we finish the proof of Claim 2.

It is also easy to see that the centroid $a$ we found above is arbitrarily close to $x$, since $a \in B(x, \delta')$ and $\delta' \leq \sqrt{\epsilon/\Lambda}$ can be arbitrarily small. $\qquad\square$

## Proof of Lemma 2:

*Proof.* First, we prove the existence of $\lim_{k \to \infty} k^{2r/d} V_{k,r,\phi}(P)$ by showing the equivalence of $\limsup_{k \to \infty}$ and $\liminf_{k \to \infty}$. But before analyzing the limit superior and limit inferior, we clarify some notations.

For a fixed $k$, let $\alpha_k^*$ be the (so far unknown) optimal set of centers for the Bregman quantization problem:

$$\alpha_k^* = \arg \inf_{\alpha \subset \mathbb{R}^d, |\alpha| = k} E_P \left[ \min_{a \in \alpha} D_\phi^r(X, a) \right], \tag{44}$$

then,

$$\inf_{\alpha \subset \mathbb{R}^d, |\alpha| = k} E_P \left[ \min_{a \in \alpha} D_\phi^r(X, a) \right] = E_P \left[ \min_{a \in \alpha_k^*} D_\phi^r(X, a) \right]. \tag{45}$$

We dice the support of $P$ into a set of small cubes $\{A_l\}$ with size $\delta$ and with edges parallel to axes, where $l$ is the index for cells and cell size $\delta$ will be determined later. We denote $s_l = \mathcal{P}(A_l)$ and conditional density on $A_l$ as $P(\cdot|A_l)$, and it is easy to have

$$P = \sum_l s_l P(\cdot|A_l). \tag{46}$$

Let $\alpha_l = \alpha \cap A_l$ (correspondingly, $\alpha_{k,l}^* = \alpha_k^* \cap A_l$) be the set of centroids that located in $A_l$ and $k_l = |\alpha_l|$ be the size of $\alpha_l$, and ratio $v_l = k_l/k$. In each cell, we choose a representative point $z_l$, which can typically be the central point of the cell.

Since Hessian matrix $\nabla^2 \phi$ is symmetric and positive definite anywhere in $\Omega$, it is always possible to diagonalize it as

$$\nabla^2 \phi = U^T \text{diag}(\lambda_1, \cdots, \lambda_d) U, \text{ with } \forall i = 1 \cdots d, \lambda_i > 0, \tag{47}$$

where $U$ is an orthogonal transformation matrix. For an arbitrary point $x \in \Omega \subseteq \mathbb{R}^d$ and $\delta_1 > 0$, define $A_{\delta_1}(x) := [x_i - \delta_1/2, x_i + \delta_1/2]^d$ as the $d$-dimensional cube with size $\delta_1$ centered at point $x \in \mathbb{R}^d$, where $x_i$ is $i$-th coordinate of $x$.

Because $\nabla^2 \phi$ is uniformly continuous on $\Omega$, we have

$$\forall \epsilon > 0, \exists \delta_1 > 0, \forall x \in \Omega, \forall p, q \in A_{\delta_1}(x) \cap \Omega, |\lambda_i(p) - \lambda_i(q)| < \epsilon. \tag{48}$$

Therefore, Hessian matrix $\nabla^2\phi$ within a cube $A_l$, with cube size $\delta_1$ or smaller, can be approximated by $\nabla^2\phi(z_l)$ with small error. Specifically,

$$\forall\epsilon > 0, \exists\delta_1 > 0, \forall(z \in A_l\cap\Omega, \mathbf{v} \in \mathbb{R}^d), \mathbf{v}^T(\nabla^2\phi(z_l)-\epsilon\mathbf{I})\mathbf{v} \le \mathbf{v}^T\nabla^2\phi(z)\mathbf{v} \le \mathbf{v}^T(\nabla^2\phi(z_l)+\epsilon\mathbf{I})\mathbf{v} \tag{49}$$

Further more, due to the uniform continuity of $\nabla^2\phi$ and compactness of $\Omega$, $\nabla^2\phi$ is bounded on $\Omega$. Specifically,

$$\forall(x \in \Omega, \mathbf{v} \in \mathbb{R}^d), \exists(M, m \in \mathbb{R}^+, M > m), \text{ s.t. } m \cdot \|\mathbf{v}\|^2 \le \mathbf{v}^T\nabla^2\phi(z)\mathbf{v} \le M \cdot \|\mathbf{v}\|^2. \tag{50}$$

On the other hand, the data density $P$ can be approximated by a "step" density function $\tilde{P}$. Suppose $P$ is absolutely continuous for now and define

$$\tilde{P} := \sum_l s_l U(A_l), \tag{51}$$

where $U(A_l)$ is the uniform density within $A_l$ and 0 anywhere else. Then,

$$\forall\epsilon > 0, \exists\delta_2 > 0, \forall z \in A_l \cap \Omega, |P(z) - \tilde{P}(z)| \le \epsilon. \tag{52}$$

Set $\delta = \min\{\delta_1, \delta_2\}$. Applying Lemma 1, we conclude that, in the limit of $k \to \infty$, each interior point $z$ of $A_l$ is assigned to a centroid $a \in A_l$, therefore the corresponding $\xi$ in Eq.(10) is also in $A_l$. Considering that the union of boundaries of $A_l$ is measure zero and applying Eq.(46), Eq.(49) and Eq.(52), we have

$$
\begin{aligned}
&\limsup_{k\to\infty} k^{2r/d}V_{k,r,\phi}(P) \\
=& \limsup_{k\to\infty} k^{2r/d}E_P\left[\min_{a\in\alpha_k^*} D_\phi^r(X, a)\right] \\
=& \limsup_{k\to\infty} k^{2r/d}E_P\left[\min_{a\in\alpha_k^*}\left(\frac{1}{2}(X-a)^T\nabla^2\phi(\xi)(X-a)\right)^r\right] \\
\le& \limsup_{k\to\infty} k^{2r/d}E_{\tilde{P}}\left[\min_{a\in\alpha_k^*}\left(\frac{1}{2}(X-a)^T\nabla^2\phi(\xi)(X-a)\right)^r\right] \\
&+\epsilon\cdot\limsup_{k\to\infty} k^{2r/d}\int_\Omega \min_{a\in\alpha_k^*}\left(\frac{1}{2}(X-a)^T\nabla^2\phi(\xi)(X-a)\right)^r dX \\
\le& \limsup_{k\to\infty} k^{2r/d}\sum_l s_l E_{U(A_l)}\left[\min_{a\in\alpha_{k,l}^*}\left(\frac{1}{2}(X-a)^T\nabla^2\phi(z_l)(X-a)\right)^r(1+\frac{\epsilon}{m})^r\right] \\
&+\epsilon\cdot\limsup_{k\to\infty} k^{2r/d}\sum_l \int_{A_l}\min_{a\in\alpha_{k,l}^*}\left(\frac{1}{2}(X-a)^T\nabla^2\phi(z_l)(X-a)\right)^r(1+\frac{\epsilon}{m})^r dX \\
=& \limsup_{k\to\infty} k^{2r/d}\sum_l s_l E_{U(A_l)}\left[\min_{a\in\alpha_{k,l}^*}\left(\frac{1}{2}(X-a)^T\nabla^2\phi(z_l)(X-a)\right)^r\right] \\
&+\epsilon\cdot\frac{r}{m}\limsup_{k\to\infty} k^{2r/d}\sum_l s_l E_{U(A_l)}\left[\min_{a\in\alpha_{k,l}^*}\left(\frac{1}{2}(X-a)^T\nabla^2\phi(z_l)(X-a)\right)^r\right] \\
&+\epsilon\cdot\limsup_{k\to\infty} k^{2r/d}\sum_l \int_{A_l}\min_{a\in\alpha_{k,l}^*}\left(\frac{1}{2}(X-a)^T\nabla^2\phi(z_l)(X-a)\right)^r dX + o(\epsilon^2) \\
:=& \limsup_{k\to\infty} k^{2r/d}\sum_l s_l E_{U(A_l)}\left[\min_{a\in\alpha_{k,l}^*}\left(\frac{1}{2}(X-a)^T\nabla^2\phi(z_l)(X-a)\right)^r\right] + \epsilon\cdot\frac{r}{m}\cdot A + \epsilon B + o(\epsilon^2)
\end{aligned}
\tag{53}
$$

The second inequality holds in the above, because

$$
\begin{aligned}
& \frac{1}{2}(X-a)^T \nabla^2 \phi(\xi)(X-a) \\
\leq \quad & \frac{1}{2}(X-a)^T \nabla^2 \phi(z_l)(X-a) + \frac{\epsilon}{2}\|X-a\|^2 \\
\leq \quad & \frac{1}{2}(X-a)^T \nabla^2 \phi(z_l)(X-a) + \frac{\epsilon}{2m}(X-a)^T \nabla^2 \phi(z_l)(X-a) \\
= \quad & (1+\frac{\epsilon}{m}) \cdot \frac{1}{2}(X-a)^T \nabla^2 \phi(z_l)(X-a).
\end{aligned}
\tag{54}
$$

After a similar argument as above, we have

$$
\begin{aligned}
& \liminf_{k \to \infty} k^{2r/d} V_{k,r,\phi}(P) \\
\geq \quad & \liminf_{k \to \infty} k^{2r/d} \sum_l s_l E_{U(A_l)} \left[ \min_{a \in \alpha_{k,l}^*} \left( \frac{1}{2}(X-a)^T \nabla^2 \phi(z_l)(X-a) \right)^r \right] \\
& -\epsilon \cdot \frac{r}{m} \cdot A - \epsilon B - o(\epsilon^2)
\end{aligned}
\tag{55}
$$

To conclude that the sequence, $\{k^{2r/d} V_{k,r,\phi}\}$, converges to a finite value, we only need to prove equivalence of the *limsup* and *liminf* appeared in the last expressions of Eq.(53) and Eq.(55), and the finiteness of $A$ and $B$.

We are going to show that the *limit* exists for the following sequence, which will indicate the agreement of the *limsup* and *liminf*.

$$
\begin{aligned}
& k^{2r/d} \sum_l s_l E_{U(A_l)} \left[ \min_{a \in \alpha_{k,l}^*} \left( \frac{1}{2}(X-a)^T \nabla^2 \phi(z_l)(X-a) \right)^r \right] \\
= \quad & k^{2r/d} \inf_{\alpha \subset \mathbb{R}^d, |\alpha|=k} \sum_l s_l E_{U(A_l)} \left[ \min_{a \in \alpha_l} \left( \frac{1}{2}(X-a)^T \nabla^2 \phi(z_l)(X-a) \right)^r \right] \\
= \quad & \inf_{\{v_l\}} \sum_l s_l v_l^{-2r/d} k_l^{2r/d} \inf_{\alpha_l} E_{U(A_l)} \left[ \min_{a \in \alpha_l} \left( \frac{1}{2}(X-a)^T \nabla^2 \phi(z_l)(X-a) \right)^r \right]
\end{aligned}
\tag{56}
$$

Within each cube $A_l$, the Hessian matrix $\nabla^2 \phi(z_l)$ is constant and positive definite. For each $A_l$, we can preform an independent linear transformation, which corresponds to rotating and stretching along some directions on $A_l$, to absorb the effect of $\nabla^2 \phi(z_l)$. After the transformation, $A_l$ changes to $B_l$, which is not necessarily again a cube, and all the points also transform accordingly. Then we have

$$
\begin{aligned}
& k^{2r/d} \sum_l s_l E_{U(A_l)} \left[ \min_{a \in \alpha_{k,l}^*} \left( \frac{1}{2}(X-a)^T \nabla^2 \phi(z_l)(X-a) \right)^r \right] \\
= \quad & \inf_{\{v_l\}} \sum_l s_l v_l^{-2r/d} k_l^{2r/d} \inf_{\beta_l} E_{U(B_l)} \left[ \min_{b \in \beta_l} \left( \frac{1}{2}(X'-b)^T (X'-b) \right)^r \right]
\end{aligned}
\tag{57}
$$

Note that, according to the results of asymptotic Euclidean quantization,

$$
k_l^{2r/d} \inf_{\beta_l} E_{U(B_l)} \left[ \min_{b \in \beta_l} \left( \frac{1}{2}(X'-b)^T (X'-b) \right)^r \right]
\tag{58}
$$

converges to positive finite values for all $l$. Furthermore, Eq.(57) is convex with respect to $v_l$'s and has bounded domain constraint by $\sum v_l = 1$. Therefore,

$$
k^{2r/d} \sum_l s_l E_{U(A_l)} \left[ \min_{a \in \alpha_{k,l}^*} \left( \frac{1}{2}(X-a)^T \nabla^2 \phi(z_l)(X-a) \right)^r \right]
\tag{59}
$$

converges to a finite value in the limit of $k \to \infty$.

Since the term $A$ is just *limsup* of Eq.(58), its finiteness is guaranteed by the above discussion. Similarly, the term $B$ is also finite. Thus, we conclude that $\lim_{k \to \infty} k^{\frac{2r}{d}} V_{k,r,\phi}(P)$ exists.

From the above discussion, we have the observation that

$$\lim_{k\to\infty} k^{\frac{2r}{d}} V_{k,r,\phi}(P) + o(\epsilon)$$

$$= \lim_{k\to\infty} k^{2r/d} \sum_l s_l E_{U(A_l)} \left[ \min_{a\in\alpha_{k,l}^*} \left( \frac{1}{2}(X-a)^T \nabla^2\phi(z_l)(X-a) \right)^r \right] + o(\epsilon)$$

$$= \lim_{k\to\infty} k^{2r/d} \sum_l s_l E_{U(A_l)} \left[ \min_{a\in\alpha_{k,l}^*} \left( \frac{1}{2}(X-a)^T \nabla^2\phi(a)(X-a) \right)^r \right] + o(\epsilon)$$

$$= \lim_{k\to\infty} k^{2r/d} \inf_{\alpha\subset\mathbb{R}^d, |\alpha|=k} E_P \left[ \min_{a\in\alpha} \left( \frac{1}{2}(X-a)^T \nabla^2\phi(a)(X-a) \right)^r \right] + o(\epsilon) \qquad (60)$$

Since $\epsilon > 0$ is arbitrary, we get the conclusion of part (b).

$\square$

**Proof of Eq.(18):**

*Proof.* Let's start from Eq.(17). Note that Hessian matrix $\nabla^2\phi(z_l)$ is positive definite, it is always possible to find a positive definite matrix $\sqrt{\nabla^2\phi(z_l)}$ such that $\sqrt{\nabla^2\phi(z_l)} \cdot \sqrt{\nabla^2\phi(z_l)} = \nabla^2\phi(z_l)$.

Since $\nabla^2\phi(z_l)$, and also $\sqrt{\nabla^2\phi(z_l)}$, is constant matrix within $A_l$, we define new coordinates such that

$$Y = \sqrt{\nabla^2\phi(z_l)}X, \quad \text{and } b = \sqrt{\nabla^2\phi(z_l)}a. \qquad (61)$$

Then the Hessian $\nabla^2\phi(z_l)$ is absorbed into new coordinates, and cube $A_l = [z_{l,i} - \delta/2, z_{l,i} + \delta/2]^d$ changes to $B_l = [\sqrt{\nabla^2\phi(z_l)}(z_{l,i} - \delta/2), \sqrt{\nabla^2\phi(z_l)}(z_{l,i} + \delta/2)]^d$. It is easy to check that volume of $A_l$ is $V_{A_l} = \delta^d$, and volume of $B_l$ is $V_{B_l} = \delta^d \det\sqrt{\nabla^2\phi(z_l)}$. Then, we have

$$Q_{r,Mh,l}(U(A_i)) = \lim_{k_l\to\infty} k_l^{2r/d} \inf_{\alpha_l,(|\alpha_l|=k_l)} \int_{A_l} \left[ \min_{a\in\alpha_l} \frac{1}{2}(X-a)^T \nabla^2\phi(z_l)(X-a) \right]^r \frac{1}{V_{A_l}} d^d X$$

$$= \lim_{k_l\to\infty} k_l^{2r/d} \inf_{\beta_l,(|\beta_l|=k_l)} \int_{B_l} \min_{b\in\beta_l} \frac{1}{2^r} \|Y-b\|^{2r} \frac{1}{V_{A_l}} \frac{1}{\det\sqrt{\nabla^2\phi(z_l)}} d^d Y$$

$$= \lim_{k_l\to\infty} k_l^{2r/d} \inf_{\beta_l,(|\beta_l|=k_l)} \int_{B_l} \min_{b\in\beta_l} \frac{1}{2^r} \|Y-b\|^{2r} \frac{1}{V_{B_l}} d^d Y$$

$$= \frac{1}{2^r} \lim_{k_l\to\infty} k_l^{2r/d} V_{k_l,2r,Eu}(U(B_l))$$

$$= \frac{1}{2^r} Q_{2r}([0,1]^d) V_{B_l}^{2r/d}$$

$$= \frac{1}{2^r} Q_{2r}([0,1]^d) \delta^{2r} [\det \nabla^2\phi(z_l)]^{r/d}. \qquad (62)$$

$\square$

**Proof of Lemma 3:**

*Proof.* Consider $s_l$ and $[\det \nabla^2\phi(z_l)]^{-r/d}$ together and apply Lemma 6.8 of Graf and Luschgy [9]. $\square$

**Proof of Theorem 1:**

*Proof.* Noting that $Q_{r,\phi}(P)$ is achieved by taking the minimum of $Q_{r,\phi}(P, \{v_l\})$ over $V = \{\{v_l\} : \sum_l v_l = 1\}$ and then taking cube size $\delta \to 0$ and appling Lemma 3 on function (19), we have

$$
\begin{aligned}
Q_{r,\phi}(P) &= \lim_{\delta \to 0} \min_{\{v_l\} \in V} Q_{r,\phi}(P, \{v_l\}) \\
&= \lim_{\delta \to 0} \frac{1}{2^r} Q_{2r}([0,1]^d) \delta^{2r} \left( \sum_l s_l^{d/(d+2r)} [\det \nabla^2 \phi(z_l)]^{r/(d+2r)} \right)^{(d+2r)/d} \\
&= \lim_{\delta \to 0} \frac{1}{2^r} Q_{2r}([0,1]^d) \left( \sum_l P(p_l)^{d/(d+2r)} [\det \nabla^2 \phi(z_l)]^{r/(d+2r)} \right)^{(d+2r)/d} \\
&= \frac{1}{2^r} Q_{2r}([0,1]^d) \left( \int P^{d/(d+2r)} (\det \nabla^2 \phi)^{r/(d+2r)} d^d X \right)^{(d+2r)/d} \\
&= \frac{1}{2^r} Q_{2r}([0,1]^d) \| P (\det \nabla^2 \phi)^{r/d} \|_{d/(d+2r)}
\end{aligned}
\tag{63}
$$

The third equality holds because $P(z_l) = s_l/\delta^d$.

Above argument is based on the assumptions that $P$ has compact support. It can be remove by applying Step 5 and 6 in proof of Theorem 6.2 in [9]. $\square$

**Lemma 4.** $\forall A \subseteq \Omega$, *if* $\mathcal{P}(A) \neq 0$, *then*

$$
(\mathcal{P}(A) Q_{r,\phi}(P(\cdot|A)))^{d/(d+2r)} = Q_{r,\phi}(P)^{d/(d+2r)} \mathcal{P}_{r,\phi}(A).
\tag{64}
$$

*Proof.* From the definition of distribution function Eq.(24), we have, for set $A$

$$
\begin{aligned}
\mathcal{P}_{r,\phi}(A) &= \int_A P_{r,\phi} d\lambda^d \\
&= \frac{\int_A P^{d/(d+2r)} (\det \nabla^2 \phi)^{r/(d+2r)} d\lambda^d}{\int P^{d/(d+2r)} (\det \nabla^2 \phi)^{r/(d+2r)} d\lambda^d}.
\end{aligned}
\tag{65}
$$

In addition, $P(\cdot|A) = P 1_A \lambda^d / \mathcal{P}(A)$, then

$$
\begin{aligned}
Q_{r,\phi}(P(\cdot|A)) &= \frac{1}{2^r} Q_r([0,1]^d) \left( \int_A P^{d/(d+2r)} (\det \nabla^2 \phi)^{r/(d+2r)} d\lambda^d \right)^{(d+2r)/d} / \mathcal{P}(A) \\
&= \frac{1}{2^r} Q_r([0,1]^d) \left( \mathcal{P}_{r,\phi}(A) \int P^{d/(d+2r)} (\det \nabla^2 \phi)^{r/(d+2r)} d\lambda^d \right)^{(d+2r)/d} / \mathcal{P}(A) \\
&= Q_{r,\phi}(P) \mathcal{P}_{r,\phi}(A)^{(d+2r)/d} / \mathcal{P}(A).
\end{aligned}
\tag{66}
$$

Therefore,

$$
(\mathcal{P}(A) Q_{r,\phi}(P(\cdot|A)))^{d/(d+2r)} = Q_{r,\phi}(P)^{d/(d+2r)} \mathcal{P}_{r,\phi}(A).
\tag{67}
$$

Thus, we conclude this lemma. $\square$

**Proof of Theorem 2:**

*Proof.* All the following proof of this theorem is parallel to that of Theorem 7.5 in [9], except that we apply Lemma 4 instead of Lemma 7.2 in [9] and that we substitute squared Euclidean distance by Bregman divergences.

Specifically, we need to prove that, in the limit of $k \to \infty$, the sequence of the discrete measure $\mathcal{P}_k$, see Eq.(5), weakly converges to $\mathcal{P}_{r,\phi}$. Suppose the limiting measure of any vaguely convergent subsequence of $\{\mathcal{P}_k\}$ is $\tilde{\mathcal{P}}$, and we are going to show that $\tilde{\mathcal{P}}$ coincides with $\mathcal{P}_{r,\phi}$.

Consider a $d$-dimensional interval $A = (b, c]$ with $b, c \in \Omega \subseteq \mathcal{R}^d$ such that $\tilde{\mathcal{P}}(\partial A) = 0$. By vague convergence, $\mathcal{P}_k(A) \to \tilde{\mathcal{P}}(A)$. Assume $0 < \mathcal{P}(A) < 1$. Since $\mathcal{P}$ and $\mathcal{P}_{r,\phi}$ are mutually absolutely continous, this is equivlent to $0 < \mathcal{P}_{r,\phi}(A) < 1$.

Before going further, we make some notations first. We denote $\alpha_k$ as the $k$-optimal set of centers for $P$. Without ambiguity, in the setting of this proof, write $A_1 = A, A_2 = \Omega - A, s_i = \mathcal{P}(A_i), v_1 = \tilde{\mathcal{P}}(A_1), v_2 = 1 - \tilde{\mathcal{P}}(A_1), \mathcal{P}_{A_i} = \mathcal{P}(\cdot|A_i), \alpha_{i,k} = \alpha_k \cap A_i$ and $k_i = |\alpha_{i,k}|$.

For $0 < \epsilon \leq \min_{i=1,2} P_{r,\phi}(A_i)$, choose $b_i, c_i \in \Omega, b < b_1 < c_1 < c, b_2 < b < c < c_2$ such that the sets $B_1 = [b_1, c_1]$ and $B_2 = [b_2, c_2]^c \cap \Omega$ satisfy $\mathcal{P}(B_i) > 0$ and

$$\mathcal{P}_{r,\phi}(B_i) \geq \mathcal{P}_{r,\phi}(A_i) - \epsilon, \quad i = 1, 2. \tag{68}$$

Then choose a finite set $\gamma_i$ on the boundary of $B_i$ so that

$$\min_{a \in \gamma_i} D_\phi(x, a) \leq \inf_{y \in A_i^c} D_\phi(x, y), \text{ for every } x \in B_i. \tag{69}$$

We have

$$\int \min_{a \in \alpha_k} D_\phi^r(x, a) d\mathcal{P}(x) = \sum_{i=1}^2 s_i \int \min_{a \in \alpha_k} D_\phi^r(x, a) d\mathcal{P}_i(x)$$

$$\geq \sum_{i=1}^2 s_i \int_{B_i} \min_{a \in \alpha_k \cup \gamma_i} D_\phi^r(x, a) d\mathcal{P}_i(x)$$

$$= \sum_{i=1}^2 s_i \int_{B_i} \min_{a \in \alpha_{i,k} \cup \gamma_i} D_\phi^r(x, a) d\mathcal{P}_i(x)$$

$$\geq \sum_{i=1}^2 s_i V_{k_i + |\gamma_i|, r, \phi}(P(\cdot|B_i)) \mathcal{P}(B_i) / \mathcal{P}(A_i). \tag{70}$$

According to Theorem 1,

$$Q_{r,\phi}(P) = \lim_{k \to \infty} k^{2r/d} \int \min_{a \in \alpha_k} D_\phi^r(x, a) d\mathcal{P}(x)$$

$$\geq \sum_{i=1}^2 s_i v_i^{-2r/d} Q_{r,\phi}(P(\cdot|B_i)) \mathcal{P}(B_i) / \mathcal{P}(A_i). \tag{71}$$

Using Lemma 4,

$$Q_{r,\phi}(P(\cdot|B_i)) \mathcal{P}(B_i) = Q_{r,\phi}(P) \mathcal{P}_{r,\phi}(B_i)^{(d+2r)/d}$$

$$\geq Q_{r,\phi}(P)(\mathcal{P}_{r,\phi}(A_i) - \epsilon)^{(d+2r)/d} \tag{72}$$

Applying both Lemma 3 and 4 and considering $0 < \epsilon \leq \min_{i=1,2} P_{r,\phi}(A_i)$, we have

$$Q_{r,\phi}(P) \geq \sum_{i=1}^2 s_i v_i^{-2r/d} Q_{r,\phi}(P) \mathcal{P}_{r,\phi}(A_i)^{(d+2r)/d} / \mathcal{P}(A_i)$$

$$= \sum_{i=1}^2 s_i v_i^{-2r/d} Q_{r,\phi}(P_i)$$

$$\geq \sum_{i=1}^2 s_i \mathcal{P}_{r,\phi}^{-2r/d} Q_{r,\phi}(P_i) = Q_{r,\phi}(P). \tag{73}$$

This directly implies $P_{r,\phi}(A_i) = v_i = \tilde{\mathcal{P}}(A_i)$. Since $A$ is arbitrary $d$-dimensional interval, we get our conclusion. $\square$