[Reviews · NeurIPS 2016]

Reviewer 1

Summary

The author(s) consider the problem of clustering using the Bregman loss function. The paper considers a dataset with size n, whose data points are coming from a given density P. The authors are interested in finding the asymptotic behavior of clustering the dataset into k groups, when k,n go to infinity. The results in the literature give the asymptotic loss and the limiting distribution of centroids when squared Euclidean loss is used. This paper extends these results when an arbitrary Bregman loss function is used.

Qualitative Assessment

The paper is well organized and the results are explained clearly. I think the paper will benefit addressing the following comments: - The paper does not fully explain the difference between the case of square Euclidean loss and an arbitrary Bregman loss. In particular, what makes the proof harder for the general case? What is the main technical difficulty in extending the results of Euclidean loss? I think providing some intuition on proof sketch will make this more clear. - The examples provided in the paper have made the results more understandable. I suggest that it would help if the authors comment on the (possible) meaning of the results. For example, what is the term det\nabla^2\varphi? Does it correspond to a certain concept? providing intuition about this will help make results more understandable and useful. - I think that the results of the paper hold for the global optimum of the clustering problem (Otherwise, the early stopping of the Lloyd’s algorithm should not be a problem as stated in section 4). From a practical standpoint it would be useful to see what we can say about the fixed points of the algorithms used in practice; e.g. Lloyd’s algorithm. Is there any difference between the squared Euclidean loss and a general Bregman loss in this sense? Providing some (at least empirical) results on this will be of interest. - There are a few typos in the paper: Line 54: LLoyd -> Lloyd. Line 148: arbitarily-> arbitrarily. Line 158: axises -> axes. Line 158: parellel -> parallel. Line 183: uniformally -> uniformly. Line 238: eough -> enough.

Confidence in this Review

3-Expert (read the paper in detail, know the area, quite certain of my opinion)


Reviewer 2

Summary

This paper provides a two-ways asymptotic analysis for quantization with Bregman divergence: after a short introduction to quantization with Bregman divergences (Section 2), an asymptotic rate for the Bregman distortion is stated (Section 3.1), then the asymptotic distribution of optimal centroids is investigated in Section 3.2. These two theoretical results are confirmed with numerical illustration (Section 4).

Qualitative Assessment

The contribution of this paper is mostly theoretical: the results may be thought of as a slight generalization of the asymptotic rates for euclidean quantization that are to be found in 'Foundation of quantization for probability distributions', S.Graf and H.Luschgy. Though the results seem natural and are likely to be true, I am concerned with some technical points (in the appendix): p.10, ll.361-362: in fact you only proved a pointwise upper bound. To go through expectations you might need a similar result to Corollary 6.7 in G&L, or use liminf, limsup and Fatou's Lemma. p.12, l.383: Q_{r,\phi}(P) is not the minimum of Q_{r,\phi}(P,{v_l}) (but is smaller). I think that it is the purpose of the sentence p.5 l.168: "It can be shown that the approximation error of Q_{r,\phi}(P) converges to zero", but the proof of this result in the euclidean case is not that trivial (second part of Step 2 in the proof of Theorem 6.2 in G&L). Anyway, I think a proof of the result at p.12 l.385 is needed. I found some typos (p.7, l.238 "eough", for instance) that can be fixed with a spell-checker, and some other that can not: p.5 l.154 'Eq (2)' should be Eq.(15) I guess, and at p.7 ll.211-213, some 't' shoul be 'z'. A further typo hunt could be necessary.

Confidence in this Review

3-Expert (read the paper in detail, know the area, quite certain of my opinion)


Reviewer 3

Summary

This paper gives asymptotic analysis of errors and centroid distributions in clustering with the Bregman divergence, which is generalization of k-means clustering.

Qualitative Assessment

This paper is clearly written and easy to follow, and it gives the first result of asymptotic properties of clustering with the Bregman divergence, which is worth to share in the ML community. However, the motivation is not clear. For example, how do these results affect to theoretical or practical design of new clustering algorithms? More precisely, how to take into account the asymptotic behavior of clustering algorithms when the number k of clusters goes to infinity when one tries to a new efficient and effective clustering algorithms or evaluate existing algorithms? Due to lack of motivation, this study has a small impact in its current form. I think this study is more appealing when it is discussed in the context of not clustering but compression or coding of continuous objects. Minor comments: There are many grammatical mistakes including: - L.54: "LLoyd's" -> "Lloyd's" - L.68: "we are will be" -> "we will be" - L.123: "variable X take values" -> "a variable X takes values" - L.271: "It easy" -> "It is easy"

Confidence in this Review

2-Confident (read it all; understood it all reasonably well)


Reviewer 4

Summary

This paper present an asymptotic result for optimal k-means clustering, when there is infinite data and as the number of clusters goes to infinity. Essentially, if we process (15), the limiting distribution is proportional to the eigenvalues of the Hessian of the generator of the divergence.

Qualitative Assessment

There are several points that need clarification, from the most important to the less important one: * One key missing point is the reason to exist of the k^{2r/d} factor which appears in (14). The only motivation is intuition, but I do not get it at all. It must be introduced rigorously. * In (47) of the supplementary information, the authors claim that the difference between eigenvalues (max - min) of the Hessian is slammer then epsilon. If I am not mistaken, epsilon in (45), epsilon can be chosen arbitrarily small. If this is the case, then (47) is wrong, even for Mahalanobis, since is enforces the Hessian to be essentially spherical. Or if it is right, then the result cannot hold for any Bregman divergence, as claimed by the authors. This is important, because if there is a mistake, this invalidates (50) (supplement), and therefore Lemma 2, which is central to the paper. * In page 7, the authors claim “In addition, we only verify r = 1 cases here, since  the Bregman clustering algorithm, which utilizes Lloyd’s method, cannot address Bregman quantization problems with r != 1”. This, I suspect, is wrong, or the authors need to make it formal. Bregman divergences are exhaustive for the population minimiser as an average (right population minimiser) or f-mean (left population minimiser). Generalisations of Bregman divergences exist that allow to satisfy additional properties that Bregman divergences would not satisfy, such as population minimisers that are invariant to rotations of the coordinate axes [1] (and references therein). It turns out that in this case, even when there is no equivalent Bregman divergence, Lloyd’s quantisation can still be used (see for example the optimisation of the left population minimiser in Total Bregmen divergences) ! Two suggestions, in decreasing order of importance: * I strongly encourage the authors to pursue drilling down Lemma 3 and one if its implication that I can see, in the finite sample / finite k regime: there seems to be a seeding mechanism in the distribution defined in (20), and I wonder whether this seeding distribution would be better than the state of the art in this field, which is basically the infamous k-means++ algorithm [2] and its extensions to various Bregman divergences and Beyond [1,3]. * Additional remark: the “Remark” before Section 4 follows the well known fact that f-means associated to the KL and IS divergences are zero-attracting. I suggest the authors also put this in context with the paper’s results: there must be a link between the properties of the f-mean associated to the divergence and the limit distribution of centroids (Section 3.2). Typos: Pg 11 supplement: existance -> existence Ref [2] is incomplete Additional reference: [1] Richard Nock, Frank Nielsen, Shun-ichi Amari: On Conformal Divergences and Their Population Minimizers. IEEE Trans. Information Theory 62(1): 527-538 (2016) [2] David Arthur, Sergei Vassilvitskii: k-means++: the advantages of careful seeding. SODA 2007: 1027-1035 [3] Richard Nock, Raphaël Canyasse, Roksana Boreli, Frank Nielsen: k-variates++: more pluses in the k-means++. ICML 2016: 145-154

Confidence in this Review

2-Confident (read it all; understood it all reasonably well)


Reviewer 5

Summary

The paper studies the asymptotic behavior of Bregman clustering (as the number of clusters k goes to infinity) in terms of the clustering cost (the quantization error). It also studies the limiting distribution of the centroids. All of this is for data given in the form of a probability distribution.

Qualitative Assessment

The paper is competently executed. My main reasons for my ratings are that I feel it's a small result that uses a fairly standard "trick" (locally look at a Bregman divergence as a kind of Mahalanobis distance) to allow deployment of similar results that have been developed for the Euclidean case (the Graf/Luschgy work). The problem itself (the limiting behavior of the clustering as k goes to infinity) is also something that I find of limited interest in a larger ML context. The authors omit a number of references on algorithms for computing bregman clustering. While I don't think that's a dealbreaker for the particular problem studied here, the omission is important and glaring. There's a large body of work on explicit algorithms for Bregman clustering, and they all make sure of a structural parameter of the divergence (roughly the degree to which its non-Euclidean) as a parameter in the running time and other resource measurements. For me, what would be interesting would be to understand how this parameter affects the results here (for example in terms of rate of convergence). The authors should acknowledge a very recent paper in AISTATS 2016 (http://jmlr.org/proceedings/papers/v51/lucic16.html) (which also has citations to more work in the area, including by Ackermann et al)

Confidence in this Review

3-Expert (read the paper in detail, know the area, quite certain of my opinion)


Reviewer 6

Summary

The paper considers Bregman clustering, a generalization of k-means clustering, when the data set is a continuous probability distribution and the number of clusters is large. The authors extend previous results for k-means to provide asymptotic quantization rates and the limiting distribution of the cluster centers. In the experimental section, the limiting distributions are experimentally verified for some choices of Bregman divergences.

Qualitative Assessment

1. I think the paper is an interesting extension of the work on k-means [9]. However, it is not clear to me how much is a simple extension of previous work and how much is a novel idea. What are your main contributions? 2. Eq.(14) suggests a rate of k^(-2r/d) for Bregman clustering whereas the previous rate of [9] for k-means is k^(-r/d). Given that the squared Euclidean distance is a Bregman divergence, how can this be reconciled? 3. The results seem to only apply to distributions that are absolutely continuous with respect to Lebesgue measure. Could this restriction be potentially lifted? If not, it might make sense to mention this caveat both in the introduction and the abstract. 4. Presentation: The content is well explained and despite the technical nature I find the paper rather easy to follow. However, there are a lot of typos that could have been fixed with a serious round of proofreading (e.g. two instead of to in L140, e*n*ough in L238). In particular, the definite article "the" seems to be missing all over the paper. 5. I believe the "n" in L124 should be a k.

Confidence in this Review

1-Less confident (might not have understood significant parts)